# Consistent Video-to-Video Transfer Using Synthetic Dataset

**Jiaxin Cheng, Tianjun Xiao & Tong He**
Amazon Web Services Shanghai AI Lab
{cjiaxin,tianjux,htong}@amazon.com

## Abstract

We introduce a novel and efficient approach for text-based video-to-video editing that eliminates the need for resource-intensive per-video-per-model finetuning. At the core of our approach is a synthetic paired video dataset tailored for video-to-video transfer tasks. Inspired by Instruct Pix2Pix's image transfer via editing instruction, we adapt this paradigm to the video domain. Extending the Prompt-to-Prompt to videos, we efficiently generate paired samples, each with an input video and its edited counterpart. Alongside this, we introduce the Long Video Sampling Correction during sampling, ensuring consistent long videos across batches. Our method surpasses current methods like Tune-A-Video, heralding substantial progress in text-based video-to-video editing and suggesting exciting avenues for further exploration and deployment. https://github.com/amazon-science/instruct-video-to-video/tree/main

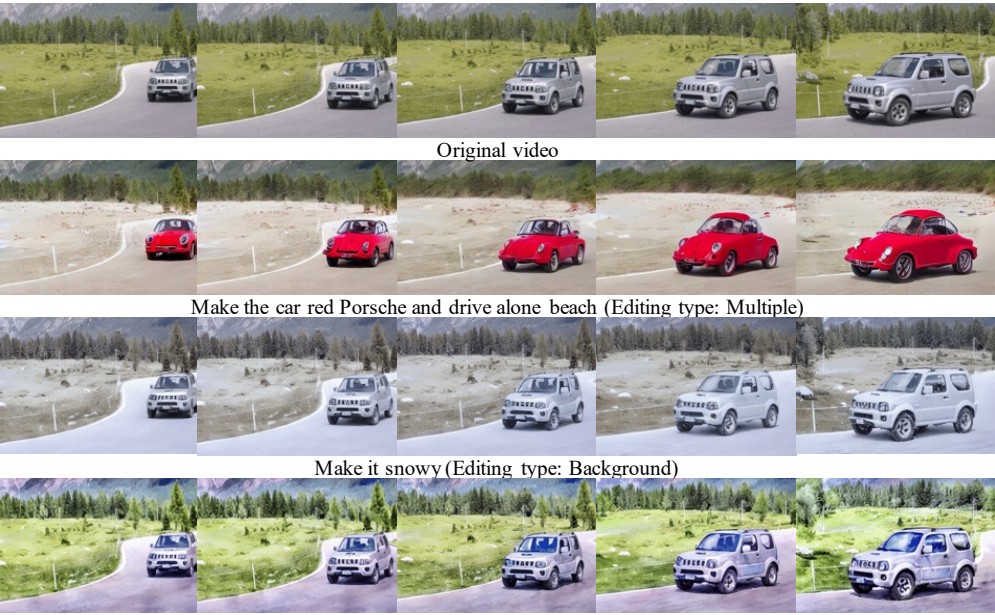

Original video

Make the car red Porsche and drive alone beach (Editing type: Multiple)

Make it snowy (Editing type: Background)

Make it watercolor (Editing type: Style)

Figure 1: InsV2V has versatile editing capabilities encompassing background, object, and stylistic changing. Our method adopts a one-model-all-video strategy, achieving comparable performance while necessitating only inference. InsV2V eliminates the need to specify prompts for both original and target videos, simplifying the process by requiring only an edit prompt, thereby enhancing intuitiveness in video editing.

# 1 INTRODUCTION

Text-based video editing Wu et al. (2022); Zhao et al. (2023); Wang et al. (2023a); Qi et al. (2023); Liu et al. (2023) has recently garnered significant interest as a versatile tool for multimedia content manipulation. However, existing approaches present several limitations that undermine their practical utility. Firstly, traditional methods typically require per-video-per-model finetuning, which imposes a considerable computational burden. Furthermore, current methods require users to describe both the original and the target video Wu et al. (2022); Zhao et al. (2023); Wang et al. (2023a); Qi et al. (2023); Liu et al. (2023). This requirement is counterintuitive, as users generally only want to specify what edits they desire, rather than providing a comprehensive description of the original content. Moreover, these methods are constrained to individual video clips; if a video is too long to fit into model, these approaches fail to ensure transfer consistency across different clips.

To overcome these limitations, we introduce a novel method with several distinctive features. Firstly, our approach offers a universal one-model-all-video transfer, freeing the process from per-video-per-model finetuning. Moreover, our model simplifies user interaction by only necessitating an intuitive editing prompt, rather than detailed descriptions of both the original and target videos, to carry out desired alterations. Secondly, we develop a synthetic dataset precisely crafted for video-to-video transfer tasks. Through rigorous pairing of text and video components, we establish an ideal training foundation for our models. Lastly, we introduce a sampling method specifically tailored for generating longer videos. By using the transferred results from preceding batches as a reference, we achieve consistent transfers across extended video sequences.

We introduce Instruct Video-to-Video (InsV2V), a diffusion-based model that enables video editing using only an editing instruction, eliminating the need for per-video-per-model tuning. This capability is inspired by Instruct Pix2Pix Brooks et al. (2023), which similarly allows for arbitrary image editing through textual instructions. A significant challenge in training such a model is the scarcity of naturally occurring paired video samples that can reflect an editing instruction. Such video pairs are virtually nonexistent in the wild, motivating us to create a synthetic dataset for training.

Our synthetic video generation pipeline builds upon a large language model (LLM) and the Prompt-to-Prompt Hertz et al. (2022) method that is initially designed for image editing tasks (Figure 2). We use an example-driven in-context learning approach to guide the LLM to produce these paired video descriptions. Additionally, we adapt the Prompt-to-Prompt (PTP) method to the video domain by substituting the image diffusion model with a video counterpart Ho et al. (2022c). This modification enables the generation of paired samples that consist of an input video and its edited version, precisely reflecting the relationships delineated by the editing prompts.

In addressing the limitations of long video editing in conventional video editing methods, we introduce Long Video Sampling Correction (LVSC). This technique mitigates challenges arising from fixed frame limitations and ensures seamless transitions between separately processed batches of a lengthy video. LVSC employs the final frames of the previous batch as a reference to guide the generation of subsequent batches, thereby maintaining visual consistency across the entire video. We also tackle issues related to global or holistic camera motion by introducing a motion compensation feature that uses optical flow. Our empirical evaluations confirm the effectiveness of LVSC and motion compensation in enhancing video quality and consistency.

# 2 RELATED WORK

**Diffusion Models** The advent of the diffusion model Sohl-Dickstein et al. (2015); Ho et al. (2020) has spurred significant advancements in the field of image generation. Over the course of just a few years, we have observed the diffusion model making groundbreaking progress in a variety of fields. This includes areas such as super-resolution Saharia et al. (2022c), colorization Saharia et al. (2022a), novel view synthesis Watson et al. (2022), style transfer Zhang et al. (2023), and 3D generation Poole et al. (2022); Tang et al. (2023); Cheng et al. (2023b). These breakthroughs have been achieved through various means. Some are attributable to the enhancements in network structures such as Latent Diffusion Models (also known as Stable Diffusion) Rombach et al. (2022), GLIDE Nichol et al. (2022), DALLE2 Ramesh et al. (2022), SDXL Podell et al. (2023), and Imagen Saharia et al. (2022b). Others are a result of improvements made in the training paradigm Nichol & Dhariwal (2021); Song & Ermon (2019); Dhariwal & Nichol (2021); Song et al. (2020b;a). Fur-

thermore, the ability to incorporate various conditions during image generation has played a crucial role. These conditions include elements such as layout Cheng et al. (2023a); Rombach et al. (2022), segmentation Avrahami et al. (2023; 2022); Balaji et al. (2022); Yang et al. (2023a), or even the use of an image as reference Mou et al. (2023); Ruiz et al. (2023); Gal et al. (2022).

**Diffusion-based Text-Guided Image Editing** Image editing is a process where we don't desire completely unconstrained generation but modifying an image under certain guidance (*i.e.* a reference image) during its generation. Various methods have been proposed to address this task. Simple zero-shot image-to-image translation methods, such as SDEdit Meng et al. (2021) performed through diffusion and denoising on reference image. Techniques that incorporate a degree of optimization, such as Imagic Kawar et al. (2023), which utilizes the concept of textual inversion Gal et al. (2022), and Null-text Inversion Mokady et al. (2023), which leverages the Prompt-to-Prompt strategy Hertz et al. (2022) to control the behavior of cross-attention in the diffusion model for editing, have also been explored. These methods can impede the speed of editing due to the necessity for per-image-per-optimization. Models like Instruct Pix2Pix Brooks et al. (2023) have been employed to achieve image editing by training on synthetic data. This approach adeptly balances editing capabilities and fidelity to the reference image.

**Diffusion-based Text-Guided Video Editing** The success of the diffusion model in image generation has been extended to video generation as well Ho et al. (2022b); Harvey et al. (2022); Blattmann et al. (2023); Mei & Patel (2023); Ho et al. (2022a), and similarly, text-guided video editing has sparked interest within the community. Techniques akin to those in image editing have found applications in video editing. For instance, Dreamix Molad et al. (2023) uses diffusion and denoising for video editing, reminiscent of the approach in SDEdit Meng et al. (2021). Strategies altering the behavior of the cross-attention layer to achieve editing, like the Prompt-to-Prompt Hertz et al. (2022), have been adopted by methods such as Vid2Vid-Zero Wang et al. (2023a), FateZero Qi et al. (2023), and Video-p2p Liu et al. (2023). Recent developments Wang et al. (2023b); Esser et al. (2023); Zhao et al. (2023) leverage certain condition modalities extracted from the original video, like depth maps or edge sketches, to condition the video generation. The related, albeit non-diffusion-based, method Text2live Bar-Tal et al. (2022) also provides valuable perspectives on video editing.

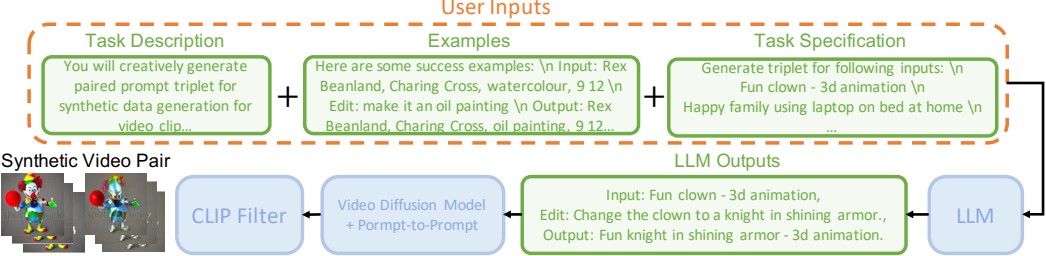

Figure 2: The pipeline for generating a synthetic dataset using a large language model, whose outputs include the prompt triplet consisting of input, edit, and edited prompts, as well as a corresponding pair of videos. Visualization of generated videos can be found in Appendix B

## 3   SYNTHETIC PAIRED VIDEO DATASET

A crucial element for training a model capable of arbitrary video-to-video transfer, as opposed to a per-video-per-model approach, lies in the availability of ample paired training data. Each pair consists of an input video and its corresponding edited version, providing the model with a diverse range of examples essential for generalized performance. However, the scarcity of naturally occurring video pairs with such correspondence poses a significant challenge to the training process.

To address this, we introduce the concept of a trade-off between initial training costs, including dataset creation, and long-term efficiency. We advocate for the use of synthetic data, which, while incurring an upfront cost, accurately maintains the required correspondence and fulfills the conditions for effective video-to-video transfer learning. The merit of this synthetic data generation approach is underscored by its potential to offset the initial investment through substantial time savings and efficiency in the subsequent inference stages. This approach contrasts with 'per-vid-

per-model' methods that necessitate repetitive fine-tuning for each new video, making our strategy both cost-effective and practical in diverse real-world applications.

The potential of synthetic data generation, well-documented in the realm of text-guided editing Brooks et al. (2023), is thereby extended to video-to-video transfer. This method allows us to construct the optimal conditions for the model to learn, offering a practical solution to the inherent obstacles associated with acquiring matching real-world video pairs.

### 3.1 DATASET CREATION:

In order to generate the synthetic dataset, we leverage the approach of Prompt-to-Prompt (PTP) Hertz et al. (2022), a proven method for producing paired samples in the field of image editing. The PTP employs both self-attention and cross-attention replacements to generate semantically aligned edited images. In self-attention, the post-softmax probability matrix of the input prompt replaces that of the edited prompt. The cross-attention replacement specifically swaps the text embedding of the edited prompt with that of the input prompt.

In the context of video-to-video transfer, we adapt PTP by substituting its underlying image diffusion models with a video diffusion model. In addition, we extend the self-attention replacement to temporal attention layers, a critical modification for maintaining structural coherence between input and edited videos. Figure 2 shows the overall pipeline for data generation. To guide the synthetic data generation, we employ a set of paired text prompts, comprising an input prompt, an edited prompt, and an edit prompt. The input prompt corresponds to the synthetic original video, while the edited prompt and the edit prompt represent the desired synthetic edited video and the specific changes to be applied on the original video respectively.

### 3.2 PROMPT SOURCES

Our synthetic dataset is constructed using paired prompts from two differentiated sources, each serving a specific purpose in the training process. The first source, LAION-IPTP, employs a fine-tuned GPT-3 model to generate prompts based on 700 manually labeled captions from the LAION-Aesthetics dataset Brooks et al. (2023). This yielded a set of 450,000 prompt pairs, of which 304,168 were utilized for synthetic video creation. While the GPT-3-based prompts offer a substantial volume of data, they originate from image captions and thus have limitations in their applicability to video generation. This led us to incorporate a second source, WebVid-MPT, which leverages video-specific captions from the WebVid 10M dataset Bain et al. (2021). Using MPT-30B Team (2023) in a zero-shot manner, we devised a set of guidelines (see Appendix A for details) to generate the needed paired prompts, adding an additional 100,000 samples to our dataset. Crucially, the WebVid-MPT source yielded a threefold increase in the success rate of generating usable samples compared to the LAION-IPTP source after sample filtering, reinforcing the need for video-specific captions in the training process. The LAION-IPTP prompt source demonstrated a success rate of 5.49% (33,421 successful generations from 608,336 attempts). The WebVid-MPT prompt source showed a higher success rate of 17.49% (34,989 successful generations from 200,000 attempts).

### 3.3 IMPLEMENTATION DETAILS AND SAMPLE SELECTION CRITERIA

**Implementation Details:** We use public available text-to-video model[1] for generating synthetic videos. Each video has 16 frames, processed over 30 DDIM Song et al. (2020a) steps by the diffusion model. The self-attention and cross-attention replacements in the Prompt-to-Prompt model terminate at a random step within the ranges of 0.3 to 0.45 and 0.6 to 0.85 respectively (out of 30 steps). The classifier-free guidance scale is a random integer value between 5 and 12.

**Sample Selection Criteria:** To ensure the quality of our synthetic dataset, we employ a CLIP-based filtering. For each prompt, generation is attempted with two random seeds, and three distinct clip scores are computed: CLIP Text Score: Evaluates the cosine similarity between each frame and the text. CLIP Frame Score: Measures the similarity between original and edited frames. CLIP Direction Score: Quantifies the similarity between the transition of original-frame-to-edited-frame and original-text-to-edited-text. These scores are obtained for each of the 16 frames and averaged.

---

[1]https://modelscope.cn/models/damo/text-to-video-synthesis/summary

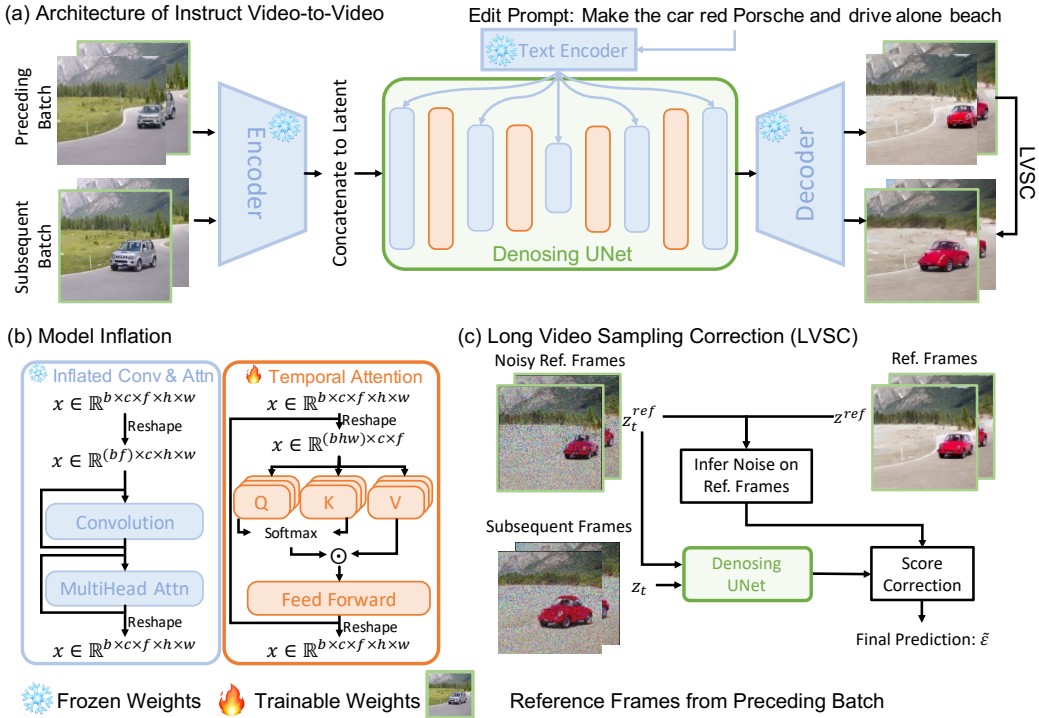

Figure 3: (a) The architecture of InsV2V. For handling long videos processed in multiple batches, our approach leverages the proposed LVSC to utilize the final frames from the preceding batch as reference frames for the subsequent batch. (b) The inflated convolutionals and attention layer, as well as temporal attention layer can handle 5D video tensors by dynamically reshaping them. (c) During each denoising iteration, the LVSC adjusts the predicted noise $\tilde{\varepsilon}_\theta(z_t)$ based on reference frames $z_t^{ref}$ prior to executing the DDIM denoising.

A sample is preserved if it meets the following conditions: CLIP Text Score > 0.2 for both original and edited videos, CLIP Direction Score > 0.2, and CLIP Frame Score > 0.5. Samples that fail to meet these criteria are discarded.

# 4 MODEL ARCHITECTURE

## 4.1 PRELIMINARIES

**Diffusion models** learn to predict content in an image by iteratively denoising an entirely random Gaussian noise. In the training process, the input image is first corrupted by the Gaussian noise, termed diffusion. The model's task is to restore a diffused noisy image to its original form. This process can be considered as the optimization of the variational lower bound on the distribution $p(x)$ of the image $x$ with a $T$ step Markovian inverse process. Various conditions $c$, such as text, images, and layouts, can be incorporated into diffusion models during the learning process. The model $\varepsilon_\theta$ we aim to train is conditioned on text and its training loss can be expressed as

$$L = \mathbb{E}_{x,\varepsilon \sim \mathcal{N}(0,1),t} \|\varepsilon - \varepsilon_\theta(x_t, t, c)\|^2 \tag{1}$$

Where $x_t$ is the result of diffusing the input image $x$ at timestep $t \in [1, T]$ using random noise $\varepsilon$. In practice, we employ the Latent Diffusion Model (LDM) Rombach et al. (2022) as our backbone model and condition it on input videos by concatenating the videos to the latent space, as illustrated in Figure 3 (a). Instead of generating images in the RGB pixel space, LDM employs a trained Vector Quantized Variational AutoEncoder (VQVAE) Esser et al. (2021) to convert images to visual codes in latent space. This allows the model to achieve better results with the same training resources.

Specifically, the image $x$ is first transformed into a latent code $z = \text{VQEnc}(x)$, and the model learns to predict $p(z)$ from random noise. In the context of the video editing, two distinct conditions exist: the editing instructions and the reference video, represented by $c_T$ and $c_V$ respectively. The model is designed to predict $p(z)$ by optimizing the following loss function:

$$L = \mathbb{E}_{\text{VQEnc}(x), \varepsilon \sim \mathcal{N}(0,1), t} \|\varepsilon - \varepsilon_\theta(z_t, t, c_V, c_T)\|^2 \tag{2}$$

## 4.2 Inflate Image-to-Image Model To Video-to-Video Model

Given the substantial similarities between image-to-image transfer and video-to-video transfer, our model utilizes a foundational pre-trained 2D image-to-image transfer diffusion model. Using foundational model simplifies training but falls short in generating consistent videos, causing noticeable jitter when sampling frames individually. Thus, we transform this image-focused model into a video-compatible one for consistent frame production. We adopt model inflation as recommended in previous studies Wu et al. (2022); Guo et al. (2023). This method modifies the single image diffusion model to produce videos. The model now accepts a 5D tensor input $x \in \mathbb{R}^{b \times c \times f \times h \times w}$. Given its architecture was designed for a 4D input, we adjust the convolutional and attention layers in the model (Figure 3 (b)). Our inflation process involves: (1) Adapting convolutional and attention layers to process a 5D tensor by reshaping it temporarily to 4D. Once processed, it's reverted to 5D. (2) Introducing temporal attention layers for frame consistency. When these layers handle a 5D tensor, they reshape it to a 3D format, enabling pixel information exchange between frames via attention.

## 4.3 Sampling

During sampling, we employ an extrapolation technique named Classifier-Free Guidance (CFG) Ho & Salimans (2022) to augment the generation quality. Given that we have two conditions, namely the conditional video and the editing prompt, we utilize the CFG method for these two conditions as proposed in Brooks et al. (2023). Specifically, for each denoising timestep, three predictions are made under different conditions: the unconditional inference $\varepsilon_\theta(z_t, \varnothing, \varnothing)$, where both the conditions are an empty string and an all-zero video; the video-conditioned prediction $\varepsilon_\theta(z_t, c_V, \varnothing)$; and the video and prompt-conditioned prediction $\varepsilon_\theta(z_t, c_V, c_T)$. Here, we omit timestep $t$ for symbolic convenience. The final prediction is an extrapolation between these three predictions with video and text classifier-free guidance scale $s_V \geq 1$ and $s_T \geq 1$.

$$\begin{aligned}
\tilde{\varepsilon}_\theta(z_t, c_V, c_T) = \ & \varepsilon_\theta(z_t, \varnothing, \varnothing) \\
& + s_V \cdot (\varepsilon_\theta(z_t, c_V, \varnothing) - \varepsilon_\theta(z_t, \varnothing, \varnothing)) \\
& + s_T \cdot (\varepsilon_\theta(z_t, c_V, c_T) - \varepsilon_\theta(z_t, c_V, \varnothing))
\end{aligned} \tag{3}$$

## 4.4 Long Video Sampling Correction for Editing Long Videos

In video editing, models often face limitations in processing extended video lengths in one go. Altering the number of input frames can compromise the model's efficacy, as frame count is usually preset during training. To manage lengthy videos, we split them into smaller batches for independent sampling. While intra-batch frame consistency is preserved, inter-batch consistency isn't guaranteed, potentially resulting in visible discontinuities at batch transition points.

To address this issue, we propose Long Video Sampling Correction (LVSC): during sampling, the results from the final $N$ frames of the previous video batch can be used as a reference to guide the generation of the next batch (Figure 3 (c)). This technique helps to maintain visual consistency across different batches. Specifically, let $z^{ref} = z_0^{prev}[:, -N:] \in \mathbb{R}^{1, N, c, h, w}$ denote the last N frames from the transfer result of the previous batch. Here, to avoid confusion with the batch size, we set the batch size to 1. The ensuing batch is the concatenation of noisy reference frames and subsequent frames $[z_t^{ref}, z_t]$. On the model's prediction $\varepsilon_\theta(z_t) := \varepsilon_\theta(z_t, t, c_V, c_T)$, we implement a score correction and the final prediction is the summation between raw prediction $\varepsilon_\theta(z_t)$ and correction term $\varepsilon_t^{ref} - \varepsilon_\theta(z_t^{ref})$, where $\varepsilon_t^{ref}$ is the closed-form inferred noise on reference frames. For notation simplicity, we use $\varepsilon(z_t^{ref})$ and $\varepsilon(z_t)$ to denote the model's predictions on reference and subsequent frames, though they are processed together by the model instead of separate inputs.

$$\varepsilon_t^{ref} = \frac{z_t^{ref} - \sqrt{\bar{\alpha}_t} z^{ref}}{\sqrt{1 - \bar{\alpha}_t}} \in \mathbb{R}^{1,N,c,h,w} \tag{4}$$

$$\tilde{\varepsilon}_\theta(z_t) = \varepsilon_\theta(z_t) + \frac{1}{N} \sum_{i=1}^{N} (\varepsilon_t^{ref}[:,i] - \varepsilon_\theta(z_t^{ref})[:,i]) \tag{5}$$

We apply averaging on the correction term when there are multiple reference frames as shown in Equations (4) and (5), where $\bar{\alpha}_t$ is the diffusion coefficient for timestep $t$ (*i.e.* $z_t = \mathcal{N}(\sqrt{\bar{\alpha}_t} z_0, (1 - \bar{\alpha}_t)I)$). In our empirical observations, we find that when the video has global or holistic camera motion, the score correction may struggle to produce consistent transfer results. To address this issue, we additionally introduce a motion compensation that leverages optical flow to establish correspondences between each pair of reference frames and the remaining frames in the batch. We then warp the score correction in accordance with this optical flow with details presented in Appendix D.

## 5 EXPERIMENTS

### 5.1 EXPERIMENTAL SETUP

**Dataset** For our evaluation, we used the Text-Guided Video Editing (TGVE) competition dataset[2]. The TGVE dataset contains 76 videos that come from three different sources: Videov, Youtube, and DAVIS Perazzi et al. (2016). Every video in the dataset comes with one original prompt that describes the video and four prompts that suggest different edits for each video. Three editing prompts pertain to modifications in *style*, *background*, or *object* within the video. Additionally, a *multiple* editing prompt is provided that may incorporate aspects of all three types of edits simultaneously.

**Metrics for Evaluation** Given that our focus is on text-based video editing, we look at three critical aspects. First, we assess whether the edited video accurately reflects the editing instructions. Second, we determine whether the edited video successfully preserves the overall structure of the original video. Finally, we consider the aesthetics of the edited video, ensuring it is free of imperfections such as jittering. Our evaluation is based on user study and automated scoring metrics. In the user study, we follow TGVE competition to ask users three key questions. The **Text Alignment** question: Which video better aligns with the provided caption? The **Structure** question: Which video better retains the structure of the input video? The **Quality** question: Aesthetically, which video is superior? These questions aim to evaluate the quality of video editing, focusing on the video's alignment with editing instructions, its preservation of the original structure, and its aesthetic integrity. For objective metrics, we incorporate PickScore Kirstain et al. (2023) that computes the average image-text alignment over all video frames and CLIP Frame (Frame Consistency) Radford et al. (2021), which measures the average cosine similarity among CLIP image embeddings across all video frames. We prefer PickScore over the CLIP text-image score since it's tailored to more closely align with human perception of image quality, which is also noticed by Podell et al. (2023).

### 5.2 BASELINE METHODS

We benchmark InsV2V against leading text-driven video editing techniques: Tune-A-Video Wu et al. (2022), Vid2Vid-Zero Wang et al. (2023a), Video-P2P Liu et al. (2023), and ControlVideo Zhao et al. (2023). Tune-A-Video has been treated as a *de facto* baseline in this domain. Vid2Vid-Zero and Video-P2P adopt the cross-attention from Prompt-to-Prompt (PTP) Hertz et al. (2022), while ControlVideo leverages ControlNet Zhang & Agrawala (2023). We test all methods for 32 frames, but PTP-based ones, due to their computational demand, are limited to 8 frames. Baselines are processed in a single batch to avoid inter-batch inconsistencies and use latent inversion Mokady et al. (2023) for structure preservation, which causes double inference time. Conversely, our method retains the video's structure more efficiently.

We also extend the comparison to include recent tuning-free video editing methods such as Token-Flow Geyer et al. (2023), Render-A-Video Yang et al. (2023b), and Pix2Video Ceylan et al. (2023).

---

[2]https://sites.google.com/view/loveucvpr23/track4

These methods, by eliminating the necessity for individual video model tuning, present a comparable benchmark to our approach. To ensure frame-to-frame consistency, these methods either adopt cross-frame attention similar to Tune-A-Video Wu et al. (2022), as seen in Ceylan et al. (2023); Yang et al. (2023b), or establish pixel-level correspondences between features and a reference key frame as in Geyer et al. (2023). This approach is effective in maintaining quality when there are minor scene changes. However, in scenarios with significant differences between the key and reference frames, these methods may experience considerable degradation in video quality. This limitation is more clearly illustrated in Figure 12 in the Appendix.

## 5.3 MODEL DETAILS

Our model is adapted from a single image editing Stable Diffusion Brooks et al. (2023) and we insert temporal attention modules after each spatial attention layers as suggested by Guo et al. (2023). Our training procedure makes use of the Adam optimizer with a learning rate set at $5 \times 10^{-5}$. The model is trained with a batch size of 512 over a span of 2,000 iterations. This training process takes approximately 30 hours to complete on four NVIDIA A10G GPUs.

During sampling, we experiment with varying hyperparameters for video classifier-free guidance (VCFG) within the choice of [1.2, 1.5, 1.8], text classifier-free guidance to 10 and video resolutions of 256 and 384. A detailed visual comparison using these differing hyperparameters can be found in the supplementary material (Appendix E). The hyperparameters combination that achieves the highest PickScore is selected as the final sampling result. Each video is processed in three distinct batches using LVSC with a fixed frame count of 16 within a batch, including reference frames from preceeding batch, and resulting in a total frame count of 32.

## 5.4 LONG VIDEO SCORE CORRECTION AND MOTION COMPENSATION

Table 1: Comparison of motion-aware MSE and CLIP frame similarity between the last frame of the preceding batch and the first frame of the subsequent batch on TGVE dataset.

| LVSC | MC | MAMSE (%) ↓ | CLIPFrame ↑ |
|------|-----|-------------|-------------|
| ✗ | ✗ | 2.02 | 0.9072 |
| ✓ | ✗ | 1.44 | 0.9093 |
| ✓ | ✓ | 1.37 | 0.9095 |

To assess performance improvement, we employ CLIP Frame Similarity and Motion-Aware Mean Squared Error (MAMSE) as evaluation metrics. Unlike traditional MSE, MAMSE accounts for frame-to-frame motion by utilizing optical flow to warp images, thereby ensuring loss computation in corresponding regions. Incorporating Long Video Score Correction (LVSC) and Motion Compensation (MC) has led to enhanced performance as reflected in Table 1. Further qualitative comparison, detailing the benefits of LVSC and MC, are provided in Appendices C and D.

## 5.5 USER STUDY

Table 2: The first two columns display automated metrics concerning CLIP frame consistency and PickScore. The final four columns pertain to a user study conducted under the TGVE protocol, where users were asked to select their preferred video when comparing the method against the TAV.

| Method | CLIPFrame | PickScore | Text Alignment | Structure | Quality | Average |
|--------|-----------|-----------|----------------|-----------|---------|---------|
| TAV* | 0.924 | 20.36 | - | - | - | - |
| CAMP* | 0.899 | 20.71 | 0.689 | 0.486 | 0.599 | 0.591 |
| T2I_HERO* | 0.923 | 20.22 | 0.531 | 0.601 | 0.564 | 0.565 |
| Vid2Vid-Zero | 0.926 | 20.35 | 0.400 | 0.357 | 0.560 | 0.439 |
| Video-P2P | 0.935 | 20.08 | 0.355 | 0.534 | 0.536 | 0.475 |
| ControlVideo | 0.930 | 20.06 | 0.328 | 0.557 | 0.560 | 0.482 |
| TokenFlow | **0.940** | 20.49 | 0.287 | 0.563 | 0.624 | 0.491 |
| Pix2Video | 0.916 | 20.12 | 0.468 | 0.529 | 0.538 | 0.511 |
| Render-A-Video | 0.909 | 19.58 | 0.326 | 0.551 | 0.525 | 0.467 |
| InsV2V (Ours) | 0.911 | **20.76** | **0.690** | **0.717** | **0.689** | **0.699** |

*: Scores from TGVE leaderboard.

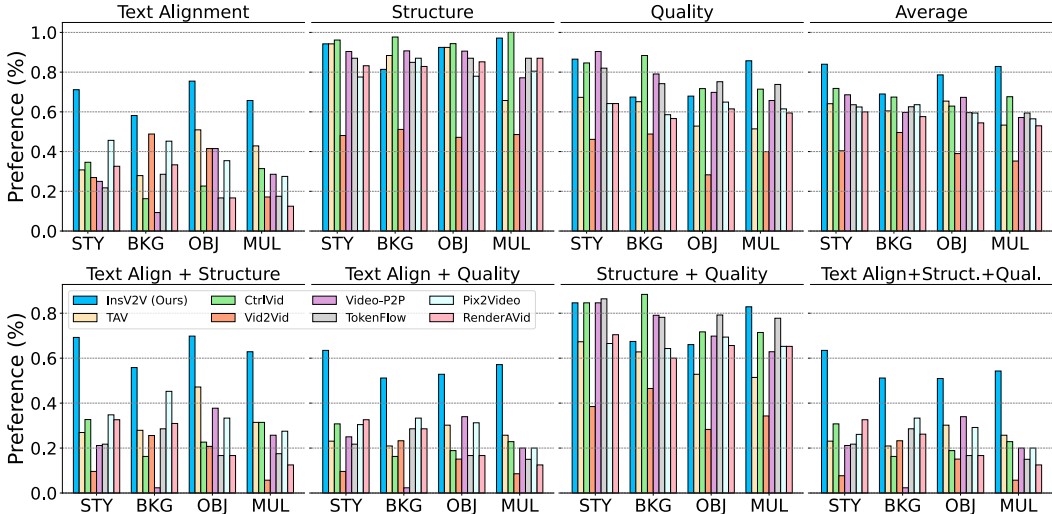

Figure 4: The abbreviations on x-axis indicate user preferences across four types of video editing within TGVE: Style, Background, Object, and Multiple. Each title specifies the evaluation metrics used for the corresponding figures. A "+" symbol signifies that the user vote meets multiple criteri. Additional qualitative results are presented in Appendix F

We conducted two separate user studies. The first followed the TGVE protocol, where we used Tune-A-Video as the baseline and compared the outcomes of our method with this well-known approach. However, we realized that this method of always asking users to choose a better option might not fully represent the reality, where both videos could either perform well or poorly. Thus, in the second user study, we compare our method with seven publicly available baselines. Instead of asking users to choose a better video, we asked them to vote for the quality of text alignment, structural preservation, and aesthetic quality for each transferred video. As evidenced by Table 2 and Figure 4, our approach excelled in all metrics except CLIP Frame similarity. We contend that CLIP Frame similarity is not an entirely apt metric because it only captures semantic similarity between individual frames, which may not be constant throughout a well-edited video due to changing scenes.

In our evaluation protocol, we additionally introduce a multi-metric assessment, capturing cases where videos satisfy multiple evaluation criteria concurrently. This composite measure addresses a key shortcoming of single metrics, which may inadequately reflect overall editing performance. For example, a high structure score might indicate that the edited video perfectly preserves the original, but such preservation could come at the expense of alignment with the editing instruction.

The results presented in Figure 4 further corroborate the advantages of our approach. While baseline methods demonstrate respectable performance when text alignment is not a prioritized criterion, they fall short when this element is incorporated into the assessment. In contrast, our method not only excels in aligning the edited video with the textual instructions but also maintains the structural integrity of the video, resulting in a high-quality output. This dual success underlines the robustness of our method in meeting the nuanced requirements of video editing tasks.

## 6 CONCLUSION

This research tackles the problem of text-based video editing. We have proposed a synthetic video generation pipeline, capable of producing paired data required for video editing. To adapt to the requirements of video editing, we have employed the model inflation technique to transform a single image diffusion model into a video diffusion model. As an innovation, we've introduced the Long Video Sampling Correction to ensure the generation of consistent long videos. Our approach is not only efficient but also highly effective. The user study conducted to evaluate the performance of our model yielded very high scores, substantiating the robustness and usability of our method in real-world applications.

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
