# OpenReview forum: "Consistent Video-to-Video Transfer Using Synthetic Dataset"
_ICLR.cc/2024/Conference — ICLR 2024 poster_

### Official Review · Reviewer_NGWA · 2023-10-31

**Soundness:** 2 fair
**Presentation:** 3 good
**Contribution:** 2 fair
**Rating:** 5
**Confidence:** 4

**Summary:**

The paper presents a method for text-driven video-to-video editing, eliminating the need for exhaustive per-video finetuning. Building on the Instruct Pix2Pix image transfer framework, the authors adapt the concept for videos, using a synthetic paired video dataset. They also introduce the Long Video Sampling Correction for consistency across longer video batches. Impressively, this approach outperforms existing techniques like Tune-A-Video, marking some advancements in the domain and opening doors for future research and application.

**Strengths:**

1. The paper crates a synthetic dataset for training instruction-based video-to-video synthesis models. This is good and could potentially benefit the community.

2. The paper is well written and easy to follow.

**Weaknesses:**

1. The paper is good at representation but some of the information is confusing. For example, in Table 1, the author claims that all baseline methods need fine-tuning and the proposed method does not need any fine-tuning. However, the proposed method also needs extra training and the cost of the dataset creation is also not reflected.

2. The paper's technical contribution seems to be incremental. The proposed long video sampling strategy seems to ve pretty similar to the sliding window operation but stated in a more formal way.

**Questions:**

Will the dataset be released?

---

> ### Author Response · Authors · 2023-11-16
>
> Thank you for your review and the rating of our paper. We appreciate your feedback and the opportunity to address your concerns.
>
> 1. Need for Fine-Tuning and Dataset Creation Costs:
>
>     * Recognizing the trade-offs between initial training costs and long-term efficiency, we appreciate your observation regarding the fine-tuning requirements and dataset creation expenses. Our method, termed 'one-model-all-vid', does require an initial training phase, a fundamental element of our approach. The primary advantage here lies in its universal applicability and enhanced efficiency during the inference phase. This is in contrast to 'per-vid-per-model' approaches that necessitate repetitive fine-tuning for each new video, as delineated in Table 1.
>     * The rationale behind our strategy is that the upfront investment in training a versatile model offers substantial benefits in terms of efficiency in subsequent inference stages. This aspect is particularly advantageous considering the varied applications in real-world scenarios. The initial costs of dataset creation and model training, we argue, are offset by these long-term efficiencies.
>     * Please refer to the revised Section 3 of our updated paper, where we explicitly discuss the trade-offs involved in creating a synthetic dataset. This elaboration aims to provide a clearer understanding of our method's rationale and its relative benefits.
>
> 2. Incremental Technical Contribution and Long Video Sampling Strategy:
>
>     * Our technical contributions include the creation of a synthetic dataset and the development of an effective method for achieving inter-batch consistency in video editing. We also trained a model capable of conducting video editing without additional control and free of per-video tuning.
>     * The Long Video Sampling Correction (LVSC) is not merely a sliding window operation. While a simple sliding window method can process video frames in overlapping segments, it does not ensure that the overlapping frames yield consistent results. The key to our LVSC lies in the algorithmic approach (detailed in Equations 4 and 5 of our paper), which is designed to align predictions across overlapping frames in different batches, ensuring consistency and coherence in the final video output.
>     * We have revised our paper to include a new illustrative figure in Appendix C, which delineates the interaction rules between preceding and subsequent batches. This addition aids in better understanding the LVSC's functionality and its distinction from a naive sliding window approach. We kindly request the reviewer to refer to the updated Appendix C for a comprehensive overview of these enhancements.
>
> 3. Dataset Release:
>
>     * We're undergoing the release process of dataset. Alongside the dataset, we also plan to make available the code necessary for reproducing the data generation, allowing for broader accessibility and replicability of our work.
>
> We hope this response addresses your concerns and clarifies the contributions and intentions of our research.

---

> > ### Comment · Reviewer_NGWA · 2023-11-22
> >
> > I thank the authors for their response. I have read the response carefully and decided to keep my original score.

---

### Official Review · Reviewer_BaKN · 2023-10-31

**Soundness:** 3 good
**Presentation:** 3 good
**Contribution:** 3 good
**Rating:** 6
**Confidence:** 4

**Summary:**

This paper proposes a novel approach for text-based video-to-video editing that eliminates the need for resource-heavy finetuning for each video and model. The authors introduce a synthetic paired video dataset tailored for video-to-video transfer tasks, taking inspiration from image transfer methods such as Instruct Pix2Pix. This method translates the Prompt-to-Prompt model to videos and efficiently generates paired samples, each consisting of an input video and its edited counterpart. They also propose Long Video Sampling Correction (LVSC), ensuring consistent long videos across batches. Their method outperforms existing methods like Tune-A-Video in terms of text-based video-to-video editing, paving new avenues for exploration and deployment.

**Strengths:**

1. The proposed method eliminates the need for per-video-per-model finetuning, potentially saving significant computational resources. The creation of a synthetic paired video dataset tailored for video-to-video transfer tasks is a novel approach that could prove beneficial for training models in this domain. The introduction of LVSC addresses the challenge of maintaining consistency in long videos across batches, a notable improvement over existing methods.
2. Sufficient and comprehensive experiments (both quantitatively and qualitatively) on the comparisons are given with prior arts and ablations of key designs. The given method gives notable quantitative improvements and its visual results faithfully follow the given instructions compared with other approaches from the supp.

**Weaknesses:**

1. The generated video shows visual appeals in the given content with different styles while presenting severe jitter in the newly added content.
2. The performance of the proposed method relies on the synthetic paired video dataset. Though it gives reasonable sampling strategies on the generated data, if this dataset doesn't closely match real-world scenarios, it may limit the model's utility.
3. The paper does not talk about potential failure cases or limitations of their approach in the main paper, which could help us better understand the proposed system.

**Questions:**

1. It would be better to quantify the difference between the generated video dataset and some reference one, e.g., computing FID between them. It may also be helpful to validate the effectiveness of the given sampling criteria.
2. How compatible is this approach with different types of video content, various editing instructions, and text-to-video generation methods?

---

> ### Author Response · Authors · 2023-11-16
>
> Thank you for your insightful review and the positive rating. We appreciate your constructive feedback and will address each point to enhance the clarity and depth of our work.
>
> 1. Jitter in Newly Added Content:
>
>     * We acknowledge the issue of jitter, particularly in newly-added content, which stems from the higher complexity in generating these elements without adequate reference data in the conditional video. To address this, a possible solution is the integration of more advanced diffusion models with better generative capacities, which may offer better handling of the complexities involved in newly added content. Additionally, enhancing the diversity of our training data could improve the model's ability to generalize, thus mitigating jitter.
>
> 2. Reliance on Synthetic Paired Video Dataset:
>
>     * The quality of our synthetic paired video dataset is indeed pivotal to our model's performance. We utilized real-world video captions to craft this dataset, aiming to mirror real-world scenarios more closely. Nevertheless, we acknowledge potential discrepancies between our dataset and actual scenarios. Future efforts could involve using text-to-video models that align more closely with real-world distributions, further bridging this gap for practical applications.
>
> 3. Discussion of Failure Cases and Limitations:
>
>     * We agree that discussing potential failure cases and limitations is crucial. Instances where our model struggles include editing targets that are partially visible or challenging for the model to semantically recognize due to size constraints. We have expanded on these failure cases in the updated appendix (see Appendix G).
>
> Regarding your questions:
>
> 1. Quantifying Differences with Reference Datasets:
>     * A quantitative comparison, like computing the Fréchet Inception Distance (FID) between our synthetic video dataset and a reference one, provides insightful data on how our dataset mirrors real-world scenarios. Beyond just assessing data realism with FID, future work can involve examining inductive biases by exploring statistical measures like the diversity in video types, their generation success rates, and variety. This will help in accurately measuring disparities with real-world scenarios using various metrics.
>     * Regarding our current CLIP-based sampling, which is heuristically driven and biased by human feedback, we note that certain prompts seem more successful in generating synthetic video pairs. Identifying whether this bias originates from the generative model or the sampling criteria is an open question. Future work can involve analyzing successful generation distributions across different prompts and their alignment with the generative model's biases, as well as assessing our sampling criteria's impact on the dataset's diversity and real-world representativeness.
>
> 2. Compatibility with Various Content, Instructions, and Methods:
>
>     * Exploring the compatibility of our method with diverse video content, editing instructions, and underlying video generation models is indeed crucial. In our research, the dataset employed for testing encompasses videos from three distinct sources, and the editing instructions cover four types. This diversity leads us to believe that our method is adept at handling a reasonable range of editing scenarios. However, our evaluation predominantly focuses on real-world videos, which are most prevalent in everyday applications. The method's compatibility with more artistic forms, such as cartoons, or semantically ambiguous videos (for instance, abstract art videos), remains less clear. We anticipate potential challenges with the latter, especially in scenarios where semantic objects are not clearly defined. Regarding editing instructions, our approach performs well when the edits do not significantly alter the video's structure, maintaining similarity between the source and target videos. However, edits that induce structural changes (like transforming a sunset scene into sunrise, where the sun's movement is reversed) pose a challenge, as our synthetic dataset does not model such scenarios effectively.
>     * Concerning Text-to-Video Generation Methods: The selection of a specific text-to-video model plays a critical role due to the inherent biases each model brings. Compiling a comprehensive synthetic dataset to account for these variances is a task that demands extensive resources. At the time of our submission, our analysis was constrained by the limited availability of text-to-video models for extensive examination. Future research directions include delving deeper into this aspect to validate the versatility of our method across a spectrum of text-to-video generation techniques.

---

> > ### Comment · Reviewer_BaKN · 2023-11-23
> >
> > I appreciate the efforts made in addressing my concerns within the responses provided; they have been well articulated and handled. However, I feel compelled to adjust my score to a 6 due to some apparent gaps concerning other reviewers' suggestions for implementing training-free methods, which unfortunately appear to remain unaddressed. I strongly encourage further exploration of these areas to holistically improve the robustness and utility of the study.

---

### Official Review · Reviewer_99Pg · 2023-11-01

**Soundness:** 3 good
**Presentation:** 2 fair
**Contribution:** 2 fair
**Rating:** 5
**Confidence:** 4

**Summary:**

This paper proposes an efficient text-based video-editing method by adapting Instruct Pix2Pix from image to video editing, eliminating the need for additional training. To enable long term video editing, a Long Video Sampling Strategy is proposed to maintain long video consistency. Experimental comparisons with other methods demonstrate the advantages of the proposed approach.

**Strengths:**

1. The paper is well-written and easy to follow
2. The paper proposed a very intersting idea for universal one-model-all-video transfer idea for vid-to-vid transfer
3. It proposed a novel synthetic dataset fo vid-to-vid transfer task.
4. The experiments were well conducted, showcasing detailed numerical indicators and a user study to evaluate the video editing method.

**Weaknesses:**

1. The proposed method in this paper lacks significant innovation. The majority of the content is derived from Instruct Pix2Pix by adapting image editing to video editing without much improvement.
2. The proposed sampling method to maintain long video consistency is a variation of inpainting sampling methods, which is also widely used in image/video generation tasks. The experimental section provides detailed numerical metrics for various evaluation indicators and user study. However, the compared baseline lacks strength in video editing tasks, there already are some video editing models based on image diffusion models, such as Pix2Video[1],  Render A Video [2], TokenFlow[3], which also do not require fine-tuning on a single video. The proposed method in this paper did not compare itself with those models mentioned.
3. From the generated video results in the provided supplementary, it seems that the proposed method does not achieve superior results.

 [1] Pix2Video: Video Editing using Image Diffusion

 [2] Rerender A Video: Zero-Shot Text-Guided Video-to-Video Translation

 [3] TokenFlow: Consistent Diffusion Features for Consistent Video Editing

**Questions:**

1. In Section 3.1, the authors mentioned that the temporal attention layers are also replaced to adapt to video editing tasks. However, in the showcased results, most videos are style transferred frame by frame. Is there any attempt to simultaneously change the style of the video and modify the motion of the video, such as transforming a person walking towards the left to walking towards the right?
2. In Section 3.2, how is the success rate calculated? Are there any other methods to improve the success rate?
3. Does using video diffusion model based methods have any advantages over using optical flow in image diffusion models in video editing tasks?

---

> ### Author Response · Authors · 2023-11-16
>
> Thank you for your review and rating. We appreciate the opportunity to address your concerns and provide further clarifications regarding our paper.
>
> 1. Perceived Lack of Significant Innovation:
>
>     * While we have built upon the high-level design philosophy of Instruct Pix2Pix, our contributions in adapting it to the video domain are significant. The introduction of Prompt-to-Prompt (PTP) for video, especially the incorporation of temporal self-attention replacement, is crucial for the success of video PTP generation, as detailed in Section 3.1. This is a substantial modification necessary for the intricacies of video data.
>     * Additionally, our work pioneers inter-batch consistency in video editing with the Long Video Sampling Correction (LVSC) and Motion Compensation (MC). These contributions are not mere extensions but are pivotal in addressing the unique challenges of video editing.
>
> 2. Sampling Method and Comparison with Baselines:
>
>     * We respectfully disagree with labeling our sampling method as just an inpainting variation. Our approach is specifically designed for diffusion models, integrating correction steps at each denoising stage's end, distinguishing it from standard inpainting methods in image or video generation.
>     * Regarding the inclusion of recent video editing models for comparison, we focused initially on methods that support extensive editing capabilities, such as significant changes in object shapes. However, we agree that incorporating recent advancements in tuning-free video editing methods is crucial for a thorough comparative analysis. Therefore, in our final camera ready version, we would like to include comparisons with the mentioned tuning-free video editing approaches [1,2,3]. We initially did not include Rerender A Video[2] in our comparison due to its primary focus on stylization editing. Nevertheless, we recognize the importance of encompassing a variety of approaches in our analysis and will aim to provide a more comprehensive comparison in the updated version of our paper.
>
> 3. Generated Video Results:
>
>     * We recommend referring to the numerical comparisons in the experimental section for a more reliable assessment of our method's performance. The supplementary videos are intended to provide additional context but may not fully represent the method's capabilities.
>
> Regarding your questions:
>
> 1. Temporal Replacement and Motion Modification:
>
>     * The temporal attention layer replacement in Section 3.1 is primarily aimed at ensuring structural similarity between the input and edited videos. This might give the appearance of frame-by-frame style transfer, but in reality, the frames are processed collectively in batches, not individually, which aligns well with our video editing framework.
>     * Regarding altering the motion in videos, such as changing the direction of a person walking, it presents an intriguing yet complex editing challenge. In scenarios like these, the structure of the input and output videos would significantly differ, which is challenging under our current approach. Our data generation pipeline is designed to produce videos with similar visual structures. While our existing model isn't equipped to handle such extensive structural changes, exploring this capability certainly presents an interesting avenue for future research.
>
> 2. Calculation and Improvement of Success Rate:
>
>     * The success rate is calculated as the ratio of successfully generated synthetic videos that pass the CLIP filter criteria (mentioned in Section 3.3) to the total number of generation attempts.
>     * To improve this rate, one approach is to source better video descriptions, as discussed in Section 3.2. Other potential methods include employing video generation models with higher capacity or refining the generation algorithm to produce more accurate and relevant synthetic videos.
>
> 3. Advantages of Video Diffusion Models Over Optical Flow-Based Methods:
>
>     * Utilizing optical flow in image diffusion models for video editing is a concept worth exploring, yet its application is not straightforward. There are two primary challenges with this approach: Firstly, when frames are processed sequentially, errors can accumulate rapidly, leading to significant inconsistencies. Secondly, occlusions in the video can create regions where warping operations struggle to maintain consistency, further complicating the editing process.
>     * In contrast, video diffusion models inherently embed consistency within their architecture, offering a learnable approach to maintaining coherence across frames. A key advantage here is that frames are processed together in a single batch, which helps in preserving the continuity and overall structure of the video more effectively than methods reliant on optical flow in image diffusion models. This integrated processing approach of video diffusion models provides a more robust and cohesive solution for video editing tasks.

---

> > ### Comment · Reviewer_99Pg · 2023-11-23
> > **Official Comment by Reviewer 99Pg**
> >
> > I thank authors for their detailed rebuttal. The explanation of the advantages of video diffusion models over flow-based methods for maintaining video consistency is insightful.
> >
> > However, after reading the rebuttal, I still feel that the paper lacks a distinct element of novelty. It primarily builds upon the Instruct Pix2Pix framework. In video, the temporal dimension is not just a consistent data channel but a critical axis that carries dynamic information. Thus, simply extending Instruct Pix2Pix to videos without addressing this temporal aspect may not fully leverage the rich motion information inherent in video.
> >
> > Secondly, I still did not see any comparison with tuning-free video editing methods [1,2,3] in the revised paper or SM. I think those are important baselines and should be compared with. Without such results, it is hard to analyze the performance of the proposed method.
> >
> > Hence, I decided to keep my original score.

---

### Official Review · Reviewer_3EH3 · 2023-11-06

**Soundness:** 3 good
**Presentation:** 3 good
**Contribution:** 3 good
**Rating:** 8
**Confidence:** 3

**Summary:**

The paper tackles the problem of text-based video editing. The proposed method, namely InsV2V, is an extension of Instruct Pix2Pix to the domain of videos. InsV2V follows the same paradigm of first generating synthetic data containing videos before and after the editing, as well as the correspoinding text. These data can then be used for training the video editing model.
The synthetic data is generated using an off-the-shelf text-to-video model. To begin with, the text prompts are obtained from existing datasets and the instructions are generated using a pretrained LLM and in-context learning. Then the source videos are generated using the text-to-video model and the target videos are generated using Prompt-to-Prompt technique. Finally, these generated video and text pairs are filtered using CLIP scores.
After the data is obtained, an video-to-video model is constructed based on a pretrained image-to-image LDM, and partially finetuned on the synthetic data.
To allow the generation of videos longer than the training data, the video-to-video model is conditioned on a few previous frames. Additionally, a score correction term based on optical flow is added to improve temporal consistency across consecutive batches of the same video.

**Strengths:**

* Text-based video editing is a difficult problem, yet the paper has managed to achieve.
* The evaluation of the method is very comprehensive -- it has included both automatic metrics as well as user studies, demonstrating state-of-the-art performance.
* The novel components proposed in the paper, namely long video score correction (LVSC) and motion compensation (MC) has significantly improved the consistency of the generated videos, as illustrated in Table 2 as well as in the supplementary material.

**Weaknesses:**

* The techniques used in the paper are not completely new -- a large portion of them has followed Instruct-Pix2Pix.
* Even with all the measures in place, the generated videos are still far from being temporally consistent.

**Questions:**

* Which LLM did you use for the in-context learning?

---

> ### Author Response · Authors · 2023-11-16
>
> Thank you for the review and the positive rating of our paper. We value your input and would like to address your points to further clarify our work's contributions and address your queries.
>
> 1. Similarity to Instruct Pix2Pix:
>
>     * Indeed, our work is inspired by the Instruct Pix2Pix framework, particularly in adopting the principle of generating paired data and tuning a model on synthetic data. However, our approach significantly extends this concept into the video domain, which involves unique challenges not present in image editing.
>     * A key innovation in our work is the application of the Prompt-to-Prompt (PTP) method to video, where we identified that temporal self-attention replacement is crucial for successful video PTP generation, as detailed in Section 3.1. This adaptation is non-trivial and vital as it ensures the paired synthetic videos retain a similar visual structure.
>     * Additionally, while previous works have focused on intra-batch consistency in video editing, our Long Video Sampling Correction (LVSC) and Motion Compensation (MC) mechanisms are pioneering techniques for maintaining inter-batch video consistency in diffusion models. These contributions represent an advancement in the field of text-guided video editing.
>
> 1. Temporal Consistency Challenges:
>
>     * Compared to existing works, our approach takes a step towards an ultimate solution for generating highly consistent videos. The implementation of Long Video Sampling Correction (LVSC) in our method represents an advancement in enhancing temporal consistency. We acknowledge that achieving absolute perfection in this domain remains a challenging and ongoing endeavor. Nonetheless, our method's progress in this area is a testament to its potential and the direction for future research to fully realize the goal of flawless temporal consistency in video editing.
>
> 1. Large Language Model (LLM) Used:
>
>     * For the in-context learning process, we utilized the MPT-30B model, as described in Section 3.2.

---

> > ### Comment · Reviewer_3EH3 · 2023-11-22
> >
> > I would like to thank the authors for the response. I keep my original rating of accept, the rationale being good result on a very challenging problem.

---

### Meta-Review · Area_Chair_nU2z · 2023-12-06

**Metareview:**

The paper receives mixed borderline reviews. On one hand, the reviewers appreciate the comprehensive quantitative and qualitative evaluations, the well-written paper, and the valuable synthetic dataset (which the authors promise to release along with the code in the rebuttal). On the other hand, the reviewers are most concerned about the limited novelty and missing comparisons to recent tuning-free video editing methods. After further checking, AC confirms the works mentioned by the reviewer released their code very close to the submission deadline of ICLR, so it should not be a main reason for rejecting the paper. The authors are strongly encouraged to include the comparisons to these methods in their camera-ready version. Regarding novelty, the authors are encouraged to emphasize the differences when adapting prompt2prompt to videos, as well as the proposed long video score correction (LVSC) in the revised manuscript.

**Justification For Why Not Higher Score:**

The novelty is quite limited. The proposed pipeline is pretty much an extension of instruct-pix2pix to videos. While there are some non-trivial design choices, they are not particularly novel.

**Justification For Why Not Lower Score:**

The main reason for reviewers to reject the paper, which is missing comparisons to recent tuning-free works, is not valid since they released their code only recently. After looking at the reviews carefully, AC believes the remaining issues are quite minor and should not be the reason to reject the paper. The proposed synthetic dataset could also be valuable to the community, which the authors promised to release.

---

### Decision · Program_Chairs · 2024-01-16

Accept (poster)